# Thrombocytopenia in murine schistosomiasis is associated with platelet uptake by liver macrophages that have a distinct activation phenotype

Joanna H. Greenman[1,2], Shinjini Chakraborty[1,2], Lucie Moss[1,2], Paul C. Armstrong[3], Paul M. Kaye[2,4], Ian S. Hitchcock[1,2]*, James P. Hewitson [1,2]*

1 Department of Biology, University of York, Heslington, York, United Kingdom, 2 York Biomedical Research Institute, University of York, Heslington, York, United Kingdom, 3 Centre for Immunobiology, Blizard Institute, Faculty of Medicine and Dentistry, Queen Mary University of London, London, United Kingdom, 4 Hull York Medical School, University of York, York, United Kingdom

* james.hewitson@york.ac.uk (JPH); ian.hitchcock@york.ac.uk (ISH)

## Abstract

Alongside their well-established role in hemostasis, platelets are key modulators of immune cell function. This is particularly the case for macrophages, as platelets can either promote or dampen macrophage activation in a context-specific manner. Whilst the role of platelets in modulating classical (M1) macrophage activation following bacterial challenge is relatively well understood, whether platelets control other macrophage responses is less clear. We investigated the role of platelets in type 2 inflammation using a mouse model of chronic schistosomiasis. Schistosome infection caused thrombocytopenia which was not fully reversed after drug-induced parasite death. Reduced platelet levels in infection were coincident with lower levels of systemic TPO and extensive liver damage caused by parasite eggs. Infection also reduced the ploidy and size (but not number) of bone marrow megakaryocytes, which was associated with reduced platelet output. We show schistosome infection accelerated platelet clearance and promoted the formation of platelet-leukocyte aggregates. This was particularly the case for liver macrophages and monocytes. Phenotypic analysis shows that platelet-associated liver macrophages had a distinct activation phenotype that included elevated expression of the alternative (M2) activation marker RELMα. Despite this, *in vitro* studies indicated that platelets do not directly promote macrophage alternative activation. Similarly, whilst *in vivo* pharmacological treatment with a TPO mimetic enhanced platelet numbers and platelet-leukocyte aggregates, this did not alter macrophage phenotype. Conversely, antibody-mediated depletion of platelets or use of platelet-deficient mice both led to extensive bleeding following infection which impacted host survival. Together, these data indicate that whilst platelets are essential to prevent excessive disease pathology in schistosomiasis, they have a more nuanced role in myeloid cell activation and type 2 immune responses.

**Data availability statement:** Data in this submission is included as a supplementary data file.

**Funding:** Funding: This work was supported by UK BBSRC White Rose Mechanistic Biology Doctoral Training Partnership award to JHG, and a UK MRC New Investigator Research Grant (MR/W018578/1), Academy of Medical Science/GCRF Springboard (SBF003/1096) and National Mouse Genetics Network (MC/PC/21043) funding to JPH. The funders had no role in study design, data collection and analysis, decision to publish, or preparation of the manuscript.

**Competing interests:** The authors have declared that no competing interests exist.

## Author summary

Platelets are the second most abundant blood cell and are best known for their role in stopping bleeding after blood vessel damage. More recent studies have revealed another important function of platelets is their ability to control immune cell activation. Here, we investigate the role of platelets in immune responses to schistosomes, parasitic worms that cause the disease schistosomiasis that affects hundreds of millions worldwide. Schistosome worms live in our blood vessels and release large numbers of eggs that must exit the blood and move through our tissues to exit the body for onward transmission. However, a large number of eggs become trapped in different organs causing inflammation and disease pathology. We find that schistosome infection reduces the numbers of platelets in the blood of laboratory mice. Platelets are taken up by liver macrophages, and whilst these macrophages have a distinct activation profile compared to other cells, platelets themselves do not cause these changes. However, platelets are essential to survive schistosomiasis due to excessive bleeding in their absence. Together, this work shows that platelets are key to surviving schistosome infection but this reflects more their role in preventing bleeding rather than controlling immune cell function.

## 1. Introduction

Platelets are best known for their role in hemostasis where they rapidly aggregate at sites of vascular injury, forming insoluble clots to prevent excessive bleeding [1]. More recently, platelets have also been shown to have key immune modulatory roles in infection and inflammation [2–6]. In this regard, platelets possess many immune-associated proteins including cytokine and chemokine receptors, pattern recognition receptors and inflammatory mediators [7–11]. Previous investigations of platelet immune-related functions have focused on pro-inflammatory type 1 responses to bacteria/bacterial PAMPs, other infections, or in non-infectious inflammatory challenges. This has revealed platelets can target several different myeloid populations including monocytes, macrophages, neutrophils, and eosinophils with context-dependent pro- or anti-inflammatory consequences [12–18]. Whether platelets modulate type 2 immune responses characteristic of allergic inflammation and anti-helminth immunity is less well understood.

Over 250 million people are infected with schistosome parasites, the causative agent of the disease schistosomiasis which has an estimated annual global health burden of >3 million disability-adjusted life years [19]. Infection occurs when free-living larvae (cercariae) released from freshwater snails penetrate human skin. Whilst initial infection stimulates a mixed Th1/Th2 cytokine response, parasite egg production from 4-5 weeks post-infection promotes a dominant type 2 inflammatory response with high levels of IL-4, IL-5 and IL-13 (weeks 6–8) [20]. In turn, the type 2 response is progressively modulated at later stages of chronic infection (between

weeks 12–20) [21]. *Schistosoma mansoni* causes intestinal schistosomiasis with parasite eggs lodging in the liver, lungs, and intestine. This leads to the formation of inflammatory granulomas containing large numbers of alternatively activated (M2) macrophages and eosinophils [22–26]. Whilst granulomatous pathology is relatively well understood, less studied is the impact of infection on hematopoiesis. This includes the development of thrombocytopenia (platelet count < 150 x $10^9$/L) in *S. mansoni* infected people, also observed in mouse models [27,28]. It is not known whether thrombocytopenia is caused by elevated platelet clearance, sequestration due schistosome-induced hepatosplenomegaly, reduced platelet production due to megakaryocyte alterations, or a combination of these [27,29,30]. Moreover, whether platelets regulate type 2 immune cell function in schistosomiasis is also not known. We address this here by testing how schistosome infection impacts platelet numbers and examining the broader role of platelets in anti-schistosome immune responses.

## 2. Materials and methods

### 2.1 Ethics statement

Ethical approval for animal work was obtained from the Animal Welfare and Ethical Review Body of University of York and procedures were performed under UK Home Office Project Licenses PFB579996, PP5712003 and P49487014. Animal experiments are reported following ARRIVE (Animal Research: Reporting of In Vivo Experiments) guidelines.

### 2.2 Mice, infections, drug cure, parasitology, in vivo IL-4 treatment

C57BL/6J wild-type (WT) mice, FcRγ$^{-/-}$ (C57BL/6N Fcer1g$^{tm1b(KOMP)Wtsi}$, supplied by University of California, Davis, USA) and Mpl$^{-/-}$ mice [31] (obtained from Prof. Warren Alexander, WEHI, Australia) were bred and maintained at the Biological Services Facility, University of York. Mice were housed in individually ventilated cages under specific pathogen-free conditions with food and water *ad libitum*. Mice were infected with parasites at 6–12 weeks of age. Anaesthetised mice were infected with 30–50 *S. mansoni* cercariae as indicated in Fig legends via percutaneous penetration of the shaved abdomen. Schistosome-infected *Biomphalaria glabrata* snails were provided by the Barrett Centre for Helminth Control (Aberystwyth University, UK). In some experiments, mice were treated with praziquantel (PZQ) at 12 weeks post-infection (oral gavage of 250mg/kg PZQ in 10% kolliphor EL for three consecutive days, both Sigma). Mice were sacrificed at indicated timepoints after infection and tissues harvested. Worm counts were determined following portal perfusion of terminally anaesthetised animals. Parasite eggs were counted under a dissecting microscope after overnight organ digestion in 4% KOH at 37°C. For schistosome egg injection experiments, parasite eggs were isolated from livers from mice infected for 7 weeks with 150 cercariae. Livers were digested overnight at 37°C with 0.2U/ml collagenase D (Roche) and 10,000U/ml polymyxin B (Sigma) with 100U/ml penicillin and 100µg/ml streptomycin (ThermoFisher). Eggs were purified by centrifugation at 450 xG for 5 minutes at room temperature with minimal break and acceleration in 33% Percoll (GE Healthcare) and 67% 0.25M sucrose, extensively washed in PBS and then counted. Eggs were rendered non-viable by storage at -20°C before use. For egg injection, mice were injected with 5000 dead eggs in PBS i.p. at d0 and i.v. at d14. For visceral leishmaniasis, mice were infected with 5x$10^7$ amastigotes of an Ethiopian strain of *Leishmani donovani* (LV9) via intravenous (i.v.) tail vein injection. For *in vivo* IL-4 treatment, recombinant IL-4 (Peprotech) was complexed with anti-mouse IL-4 mAb (clone 11B11, Ultra-LEAF Biolegend) for 30min on ice. Each mouse received 1µg rIL-4: 5µg anti-IL-4 ("IL-4 complex, IL-4c") or PBS control by i.p. injection on days 0, 2, 4, 7 and 9 with platelet parameters assessed at day 10.

### 2.3 *In vivo* platelet tracking

Mice were injected i.v. with 0.05µg/g body weight of anti-GPIb-V-IX DyLight 649 (Emfret X649) in 200µl phosphate buffered saline (PBS). Saphenous or tail bleeds were carried out at 1, 24, and 48 hours after injection and 5µl of blood was collected into 50µl acid-citrate-dextrose (ACD) buffer (13.2g/L sodium citrate dihydrate, 4.8g/L anhydrous sodium citrate, 14.7g/L D-Glucose, pH 7.4). Samples were stained with 0.2µg CD41-phycoerythrin (PE) (MwReg30, Biolegend) for 30min and diluted in 1x PBS for immediate flow cytometry analysis.

## 2.4  *In vivo* modulation of platelet levels

For acute platelet depletion, mice were injected intraperitoneally (i.p.) with 0.2mg/kg low endotoxin/azide-free (LEAF) anti-mouse CD41 (MwReg30) or IgG1κ control (both Biolegend) and harvested 12–24hrs later. For sustained platelet depletion, naive or *S. mansoni* infected mice (8 weeks post-infection) were given 0.2mg/kg antibodies as above three times a week for 1 week and then 0.4mg/kg for a further week (3x injections) based on [32]. To expand platelet numbers, mice were treated either with recombinant human TPO (10µg per mouse, 3x per week at weeks 8 and 9 post-infection, gift from Zymogenetics, USA) or romiplostim (nPlate, 2.2µg per mouse, 1x per week between week 6–9, gift from Dr David Allsup, Hull York Medical School, UK). Control mice received PBS alone.

## 2.5  Tissues and cell sorting

All centrifugation steps were carried out at 450 xG, (4°C, 5min) unless stated. Spleens were passed through 70µm cell strainers in FACS buffer (PBS with 0.5% BSA$_{(w/v)}$ and 0.05% azide$_{(v/v)}$). Livers were finely chopped in Hank's balanced salt solution (HBSS) (HyClone), digested for 45min at 37°C with 160U/ml DNase and 0.8U/ml Liberase TL (Roche Diagnostics), and the reaction stopped with 100mM EDTA and complete Dulbecco's Modified Eagle Medium (cDMEM) (Gibco) (DMEM with 10%$_{(v/v)}$ FCS, 2mM L-glutamine, 100U/ml penicillin, 100µg/ml streptomycin). Cells were passed through 100µm cell strainers, centrifuged twice, re-suspended in 8ml of 33% isotonic Percoll (GE Healthcare), and live cell pellets recovered after centrifugation (700 xG, RT, 12min with minimal acceleration or brake). Blood samples were collected into EDTA-coated tubes (Microvette CB300 EDTA; Sarstedt) and complete blood cell count analysis performed on a scil Vet abc Plus+ blood counter (Scil Animal Care Company). For blood immune cell flow cytometry, blood was collected into FACS buffer with 10mM EDTA and red blood cells lysed (5min RT) with Ammonium-Chloride-Potassium (ACK) buffer (BioWhittaker). To purify liver macrophages, liver single cell suspensions were Fc blocked with anti-CD16/32 (Clone 93, Biolegend), stained with biotinylated anti-CD64 (Clone X54-5/7.1, Biolegend, 15min) and anti-biotin MicroBeads (Miltenyi Biotec, 15min) with washes in MACS buffer (PBS, 0.5% BSA$_{(w/v)}$, 2mM EDTA), and positively selected using LS columns (Miltenyi Biotec). Enriched CD64$^+$ cells were stained with antibodies (S1 Table) and sorted as CD11b$^+$ SiglecF$^-$ Ly6G$^-$ F4/80$^+$ cells, and in some experiments additionally as CD41$^+$ *vs.* CD41$^-$ on a Beckman Coulter MoFlo Astrios. Serum TPO was quantified by ELISA (R&D systems) according to manufacturer's instructions.

## 2.6  Flow cytometry

For intracellular cytokine staining, 1-4x10$^6$ cells/well were stimulated with 1µg/ml ionomycin, 0.5µg/ml PMA and 10µg/ml Brefeldin A (all Sigma) in cDMEM for 4hrs at 37°C. Cells were washed with 1x PBS, stained with live-dead Zombie Aqua (Biolegend), Fc blocked with 100µg/ml rat IgG (Sigma), and stained with fluorescently conjugated antibodies (S1 Table). For intracellular staining (S2 Table), cells were fixed/permed with Foxp3 Transcription Factor Fixation/permeabilisation (eBiosciences, RELMα, iNOS, Ym-1) or IC Fixation/permeabilisation buffer (BD, intracellular cytokines) with paired perm/wash buffers. For flow cytometry of megakaryocytes (MK), BM was recovered from the femurs of naive and infected mice by centrifugation into ice-cold PBS (2500 xG, 40s), filtered through 100µm cell strainers and red blood cells lysed. For MK phenotyping, cells were stained with live/dead Zombie Aqua, CD41-PE and CD53-FITC, fixed and permeabilised with Foxp3 Fixation/permeabilization as above and then stained for intracellular ARNTL, LSP1 or MYLK4 then anti-rabbit secondary detection antibodies (S2 Table). For MK ploidy analysis, cells were fixed in 150µl ice-cold ethanol (30min, 4°C), washed in PBS and re-suspended in 100µl MACS buffer with 0.2µg CD41-APC, 100µg/ml propidium iodide (Biolegend) and 1mg/ml RNase A (Sigma) for 40min on ice. Cells were washed and immediately analysed by flow cytometry. All flow cytometry was carried out on a BD-LSRFortessa X-20 flow cytometer and analysed with FlowJo V10.6.1.

## 2.7 Immunofluorescence imaging and histology

Liver and spleen samples were frozen in OCT (CellPath). Femurs were fixed overnight in 4%$_{(v/v)}$ PFA/PBS, decalcified in 10%$_{(v/v)}$ EDTA pH 7.5/PBS for 2 days, cryopreserved in 20%$_{(w/v)}$ sucrose/PBS overnight (all incubations 4°C) and frozen in OCT. Tissue cryosections (8–10μm) were cut on a Leica CM 3050S cryostat and transferred to SuperFrost Plus slides. Sections were fixed in ice-cold acetone (5min) and subsequent steps carried out at RT. Sections were washed in 0.5%$_{(w/v)}$ BSA/PBSwash buffer, blocked with wash buffer supplemented with 5%$_{(v/v)}$ rat serum (30min), avidin-biotin blocked (Avidin/Biotin Blocking Solution, ThermoFisher) and washed x3 in wash buffer. Sections were incubated with primary antibodies (S3 Table), washed twice with wash buffer, incubated with fluorochrome-conjugated antibodies/streptavidin for 45min (S1 Table), washed, stained for 5min with 1μg/ml 4′,6-diamidino-2-phenylindole (DAPI, Sigma) and mounted with Pro-Long Gold (ThermoFisher). For single cell suspensions, cells were stained as above, fixed in 4%$_{(v/v)}$ PFA/PBS, stained with DAPI and cytospun onto Superfrost plus slides (Thermo scientific) using a Thermo Shandon Cytospin 4 (500 rpm, 5min). Imaging was carried out using a Zeiss LSM780 confocal microscope and analysed on Zen (Zeiss) and Fiji ImageJ software. Granuloma coverage and size was determined in formalin-fixed paraffin wax embedded liver sections (10μm) stained with Masson's trichome (Sigma) and imaged on Zeiss AxioScan.Z1 slide scanner.

## 2.8 Bone marrow macrophages (BMMϕ) differentiation and platelet co-culture

Conditioned L929 media was generated by culture of L929 cells (gift from Dr. Elmarie Myburgh, University of York) in cDMEM. Bone marrow (BM) cells were flushed from C57BL/6 femurs and tibias, and 5x10$^6$ cells seeded in 10 cm dishes with macrophage media (10ml of 30%$_{(v/v)}$ L929 supernatant in fresh cDMEM). Media was replaced on day 3 and transferred to cDMEM alone on day 6. BMMϕ were harvested on day 7 for downstream assays. To obtain platelets, blood was collected from terminally anaesthetised donor mice by cardiac puncture into ACD buffer coated tubes. An equal volume of wash buffer (2%$_{(v/v)}$ FCS in PBS) was added, platelet-rich plasma generated by centrifugation (60 xG, 7min, RT), platelets pelleted (240 xG, 10min, RT) and counted (Beckman Coulter Z1-D Dual Cell/Particle Counter). BMMϕ were seeded in 24-well plates and allowed to adhere before platelets were added (1x10$^6$ BMMϕ: 1x10$^8$ platelets, i.e., 1:100 ratio). Plates were pulse-centrifuged to promote platelet settling and incubated at 37°C. For treatment of BMMϕ with schistosome egg antigen (SEA), SEA was produced from frozen eggs crushed in PBS on ice using a glass homogeniser and then centrifuged at 10,000 xG for 30min at 4°C. Protein concentration was determined by BCA assay (Pierce) and cells were treated with 10μg/ml SEA.

## 2.9 HSC colony formation assays

BM was flushed from femurs and tibias of naive and schistosome-infected mice. For lineage depletion, 1x10$^8$ cells/ml were incubated with 20μl/ml EasySep mouse Hematopoietic Progenitor Cell Isolation Cocktail (15min, 4°C, StemCell technologies), RapidSpheres (50μl/ml, 15min, 4°C), and enriched using EasySep magnet units. Enriched samples were further stained and cells sorted as 7AAD⁻ CD45⁺ Sca1⁺ CD48⁻ CD150⁺ EPCR⁺ (EPCR⁺ hematopoietic stem cells, ESLAM HSC [33]). 100 ESLAM cells/well were cultured in MethoCult GF M3434 (StemCell Technologies) in 6-well plates -/ + 10ng/ml recombinant murine IL-4 (Peprotech) for 10 days at 37°C. Plates were imaged using STEMvision (StemCell Technologies) and colonies picked for flow cytometry analysis.

## 2.10 RNA isolation and quantitative reverse transcription PCR (qRT-PCR)

Liver tissue samples (~5mm$^3$) in QIAzol were disaggregated with TissueLyser metal beads homogenisation (QIAGEN). BMMϕ cells were washed with 1x PBS before QIAzol extraction. RNA was extracted using miRNeasy RNA extraction kits (QIAGEN) according to manufacturer's instructions. Superscript III (ThermoFisher) and random hexamer primers (Promega) were used for cDNA production. Gene expression was measured with Fast SYBR Green Master Mix

(ThermoFisher) or Taqman reagents (S4 Table) on a StepOnePlus Real Time PCR System (ThermoFisher). Transcript levels of genes were calculated using the ΔΔCT method and normalised to *Hprt* and *U6* expression [34].

### 2.11 Statistical analysis

Statistical tests were carried out using Graphpad Prism v10. Data was tested for normality and statistical significance determined using *t*-test, Mann-Whitney U test, ANOVA or Kruskal-Wallis as appropriate to the data. Experimental mice were randomly assigned to experimental groups. Blinding of researchers to experimental groups was not possible given overt infection-induced pathology. Some Figs were created in BioRender licenced to IS Hitchcock. Raw data is included in the supplementary files.

## 3. Results

### 3.1 Chronic murine schistosomiasis induces thrombocytopenia that does not fully recover after drug-induced parasite death

Thrombocytopenia can be caused by multiple different types of infection or non-infectious inflammation [35–37]. We confirmed that chronic murine schistosomiasis similarly causes a near 2-fold reduction in platelets (Fig 1A) alongside increased mean platelet volume (Fig 1B) at 12 weeks post-infection. We next asked whether this is sustained following treatment with the chemotherapeutic drug praziquantel (PZQ) which kills adult worms [38]. Mice were infected for 12 weeks to establish robust platelet changes then treated with PZQ and assessed 4 weeks later. Whilst PZQ kills worms and partially reverses infection-induced hepatosplenomegaly (S1A-S1C Fig), platelet levels in PZQ treated mice remained significantly lower than naive controls and MPV was still elevated (Fig 1C-1D). Notably however, more PZQ-treated mice trended towards increased platelet number and reduced platelet size between weeks 12 (pre-PZQ treatment) and 16 (post-PZQ treatment) compared to control infected mice (S1D-S1E Fig). There was a significant negative correlation between platelet numbers and MPV in both infected and infected/cured mice, i.e., mice with lower platelet counts had higher MPV (S1F Fig). Together, this shows infection-induced platelet changes do not fully recover 4 weeks after parasite clearance. We also tested whether injection of schistosome eggs in the absence of live infection is sufficient to alter platelet parameters. Injection of eggs on day 0 and day 14 did not cause any reduction in platelet levels at d21 but we

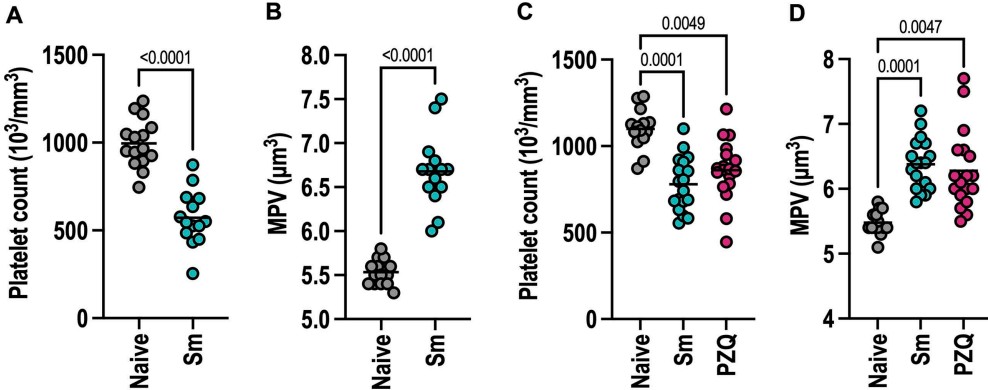

**Fig 1. Chronic murine schistosomiasis induces thrombocytopenia that does not fully recover after curative PZQ treatment.** Female C57BL/6 mice were infected with 40-50 *S. mansoni* cercariae. (A) blood platelet count and (B) mean platelet volume (MPV) were determined at 12 weeks post-infection together with naive control mice. **(C)** Platelet counts and **(D)** MPV in mice treated -/+PZQ at 12 weeks post-infection and then bled 4 weeks later (16 weeks post-infection). Data pooled from 3 independent experiments with n=4-7 per group and significance determined by unpaired *t*-test (A-B) or ANOVA with Tukey post-hoc test **(C-D)**.

did detect a modest increase in MPV (S1G-S1H Fig). This contrasts with a recent study showing egg injection induces thrombocytopenia 3 days after delivery [39]. Our results are consistent with egg injection causing only transient changes to platelets.

## 3.2 Schistosome infection does not impair megakaryopoiesis

To investigate the mechanisms of schistosome-induced thrombocytopenia, we tested whether thrombopoiesis is impaired. The main cytokine responsible for platelet production is thrombopoietin (TPO) which promotes the differentiation, maturation and survival of megakaryocytes (MK) [40,41]. TPO is produced primarily by hepatocytes in the liver, an organ which is a major site of inflammation in *S. mansoni* infection (S1B Fig). Confocal analysis of liver sections revealed schistosome egg-induced granulomas are devoid of TPO+ hepatocytes (Fig 2A). To determine whether this occurs in other infections that cause liver inflammation, we assessed TPO expression in mice infected with the protozoan parasite *Leishmania donovani* which also induces thrombocytopenia [37] (S2A Fig). Consistent with *S. mansoni* infection, *L. donovani* granulomas similarly lack TPO+ expression (Fig 2A). As infection-induced loss of TPO+ hepatocytes may be counteracted by increased liver size (S1B Fig), we assessed whether circulating TPO protein levels were altered and found both *S. mansoni* and *L. donovani* infection led to marked reductions (Fig 2B). Notably, this was still the case even 4 weeks after PZQ-mediated schistosome cure. *Thpo* message was also reduced in the livers of schistosome infected mice (S2B Fig). Despite reduced systemic TPO levels, nucleated CD41+ MK were detected in the liver (Fig 2A) of both *S. mansoni* and *L. donovani* infected mice suggestive of extramedullary hematopoiesis (Fig 2A and 2C), consistent with other studies [42]. Whilst bone marrow (BM) MK numbers were unchanged (Fig 2D-2E), schistosome infection caused a modest but significant reduction in MK size, whereas this was increased in *L. donovani* infection as previously noted (S2C Fig) [37]. We also observed schistosome (but not *L. donovani*) infection led to marked reductions in BM CD68+ macrophages (Fig 2D and S2D Fig). We next explored whether schistosome infection skews BM MK ploidy. Sun *et al.* (2021) identified three functionally and phenotypically distinct MK populations: immune/inflammatory (2-8N), HSC-niche maintenance (8-32N) and platelet generating (8-32N) [43]. Given the reduction in MK size following schistosome infection, we investigated whether this reflected endomitotic changes. MK ploidy analysis (Fig 2F) revealed infection-induced increases in 4N low-ploidy MK (immune) but reduction in 8-16N higher-ploidy cells (HSC-niche and platelet production). Consistent with this, an increased proportion of MK in schistosome infection were LSP1+ which is associated with the immune subset, although CD53 expression was unchanged (S2E-S2F Fig) [43]. In contrast, MYLK4 (HSC-niche) and ARNTL (platelet production) expression by MK did not change after infection. Finally, we tested whether schistosome infection affects the differentiation of platelet-producing cells. Colony formation assays with purified hematopoietic stem cells (HSC) from naive and schistosome-infected mice led to the production of similar numbers of colonies, and there was no difference in CD41+ expression suggesting prior infection does not alter MK lineage bias (S2G-S2I Fig). In contrast, exogenous IL-4 reduced colony number and strongly inhibited CD41+ cell differentiation, consistent with earlier reports [44,45]. This indicates that whilst IL-4 inhibits megakaryopoiesis, this either does not occur in schistosomiasis or the impact of infection-induced IL-4 is only transient. Finally, we tested whether IL-4 directly induces thrombocytopenia *in vivo* by injection of IL-4 complex (IL-4c, recombinant IL-4 bound to anti-IL-4 mAb). Consistent with previous reports [46], short-term IL-4c treatment caused a modest decrease in platelet numbers and elevated MPV (S2J-S2K Fig).

## 3.3 Schistosome infection enhances platelet clearance

As chronic schistosomiasis reduced circulating TPO levels but not MK numbers, we next asked whether platelet clearance is enhanced. Naive, schistosome or *L. donovani* infected mice were injected with a fluorescently conjugated antibody derivative (X649) that binds the GPIbβ subunit of the platelet/megakaryocyte-specific GPIb-V-IX complex (von Willebrand factor receptor) (Fig 3A). This specifically labels circulating platelets *in vivo* without reducing platelet counts allowing the measurement of platelet lifespan [47]. Longitudinal bleeds at 1 hour, 24 hours and 48 hours post-platelet

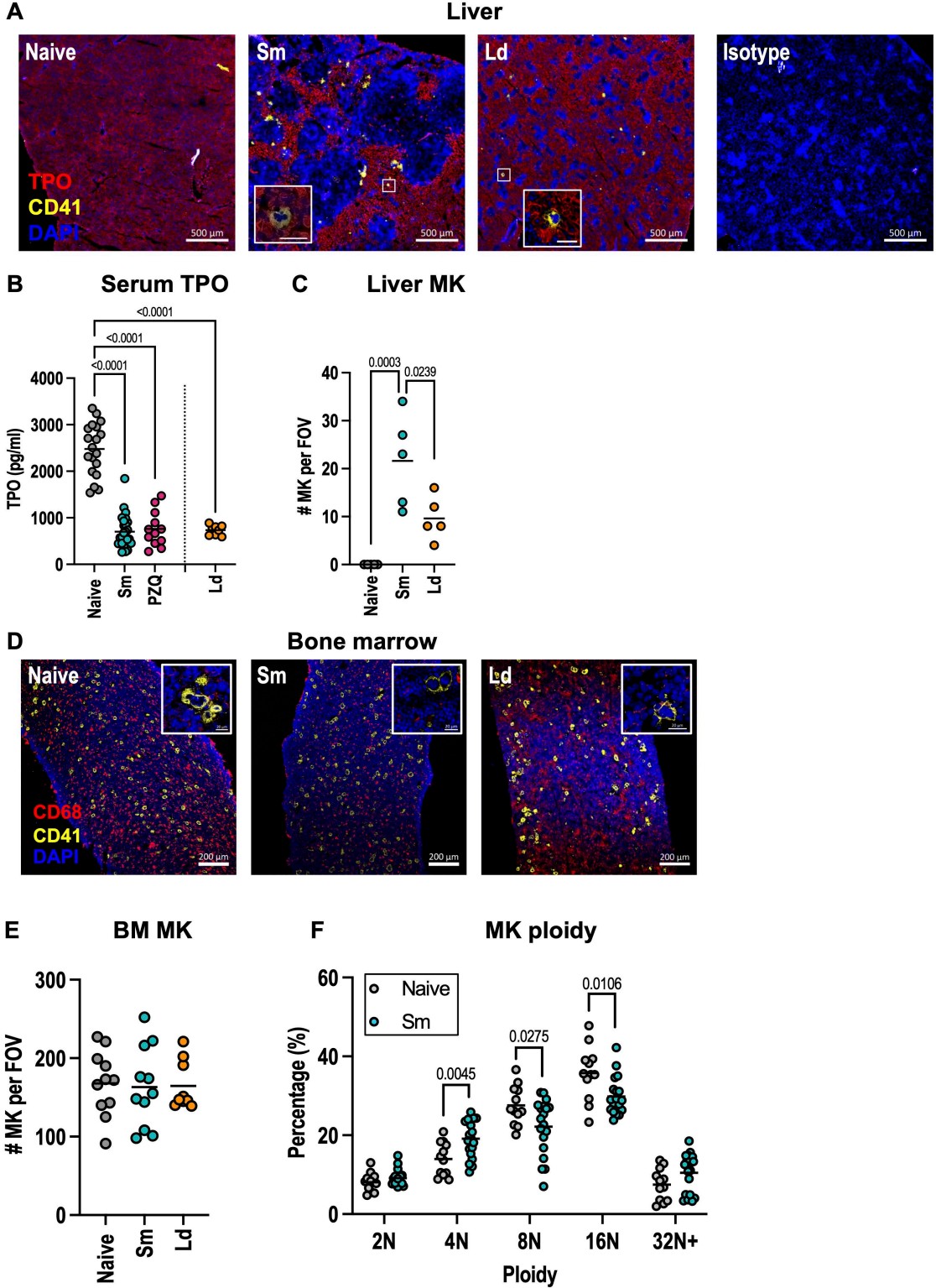

**Fig 2. Schistosome infection does not impair megakaryopoiesis.** (A and C-F) Female C57BL/6 mice were infected with 40 *S. mansoni* (*Sm*) cercariae for 10 weeks or with *L. donovani* (*Ld*) for 4 weeks. **(B)** Female C57BL/6 mice were infected with 40 *S. mansoni* (*Sm*) cercariae for 16 weeks or given PZQ at week 12 and allowed to recover for 4 weeks. **(A)** Representative liver sections stained for TPO (red), CD41 (yellow) or nuclei (DAPI, blue).

Isotype control staining for CD41 and TPO also shown. (B) serum TPO protein levels. **(C)** Liver MK quantification per field of view (FOV) with MK defined as nucleated CD41⁺ cells. **(D)** Representative bone sections stained for CD68 (macrophages, red), CD41 (MK, yellow) and DAPI nuclei (blue). **(E)** BM MK quantification per FOV. Higher magnification inserts are shown for (A) and **(D)**. **(F)** BM MK ploidy. Data is (A-B) from 2 or 4 experiments (n = 4-8 per group), (C) from a single experiment (n = 5 per group), (D-E) from 2 experiments (n = 5-6 per group), and (F) pooled from 3 experiments (n = 4-7 per group). Significance determined by ANOVA with Tukey post-hoc test (B, C, E) or unpaired *t*-test **(F)**.

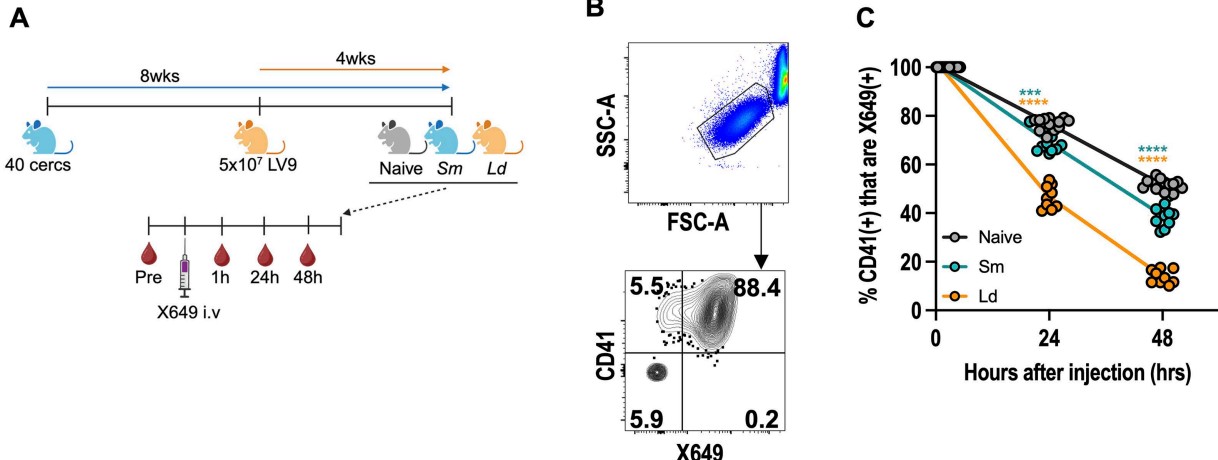

**Fig 3. Schistosome infection enhances platelet clearance. (A)** Experimental setup showing female C57BL/6 mice infected with for 12 weeks with *S. mansoni* (Sm) or 4 weeks with *L. donovani* (Ld) then injected with platelet-labelling reagent (X649) and X649-labelled platelets quantified at 1, 24 and 48 hours post-injection. Mice were also bled before X649 injection as a negative control **(B)** Representative flow cytometry of platelets from naive mice at 1 hour post-injection with additional *ex vivo* co-staining for CD41. **(C)** Quantification of X649-labelled CD41⁺ platelets in naive, schistosome and leishmania-infected mice. Data is normalised to 1hr timepoint and is pooled from 2 experiments with n = 5 per group. Significance determined by ANOVA with Tukey post-hoc test comparing Sm (blue) and Ld (orange) to naïve mice (grey) at each timepoint. Fig 3A created with BioRender.

labelling revealed schistosome infection caused significantly faster loss of labelled platelets compared to naive animals, and this was further enhanced in *L. donovani* infection (Fig 3B-3C, S3A Fig). Together, this shows both schistosome and *L. donovani* infection promote thrombocytopenia in part through accelerated platelet clearance. This does not depend on antibody-mediated uptake of opsonised platelets as schistosome-induced thrombocytopenia is maintained in FcRγ⁻/⁻ mice (S3B Fig), which are in the main resistant to antibody-mediated immune thrombocytopenia (S3C Fig).

### 3.4 Infection promotes platelet-myeloid cell interactions

In addition to revealing platelet clearance dynamics, *in vivo* X649 platelet labelling allows identification of immune cells that are interacting with platelets (i.e., platelet-leukocyte aggregates, PLA), potentially mediating their clearance and/or receiving platelet immunomodulatory signals [48]. We hypothesised CD45⁺ X649⁺ leukocytes also positive for cell surface CD41 (platelet glycoprotein IIb) represent immune cells with surface-bound platelets (i.e., X649⁺ CD41⁺), whereas cells that internalise platelets would be X649⁺ CD41⁻ (Fig 4A). In support of this, *in vitro* time course studies and fluorescence microscopy of BM-derived macrophages cultured with X649 labelled platelets showed progressive loss of surface CD41 stain as X649⁺ platelets were internalised (Fig 4B, S4A-S4B Fig). Analysis of CD45⁺ immune cells in the blood, spleen and liver of naive and schistosome infected mice 48 hours after X649 injection revealed the greatest proportion of X649⁺ cells were found in the blood, with smaller proportions in spleen and liver (Fig 4C). The majority of X649⁺ cells in the blood were also positive for surface CD41 suggesting non-internalised platelets and this was the case in both naive and schistosome infected mice (Fig 4D, S4C Fig). Whilst Ly6C⁻ non-classical monocytes had the highest percentage of X649⁺ events,

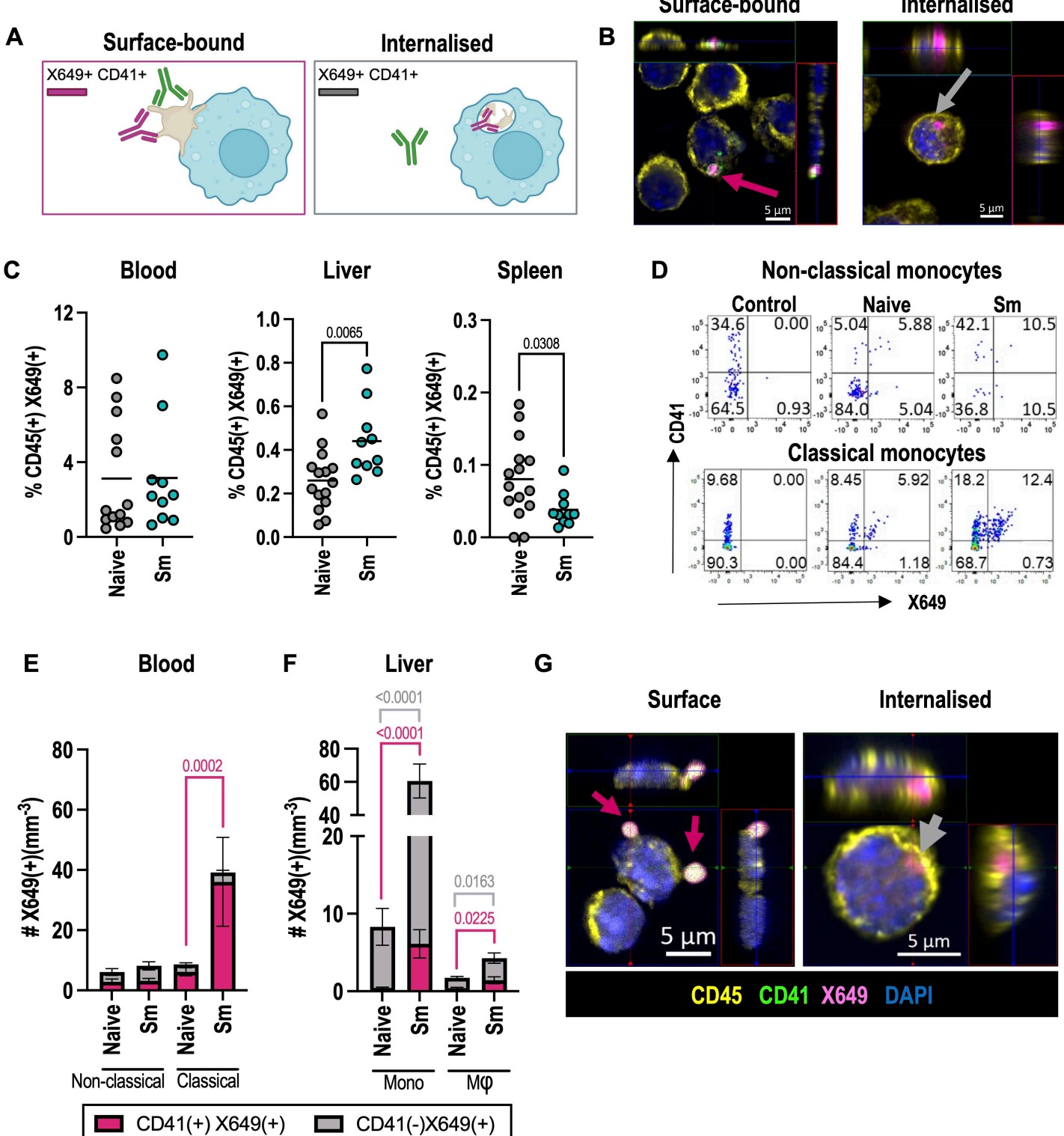

**Fig 4. Schistosome infection promotes myeloid cell-platelet interactions. (A)** Model of platelet-immune cell interactions showing cells with surface-bound platelets as X649⁺ CD41⁺ and internalised platelets as X649⁺ CD41⁻. **(B)** BMMφ were co-cultured with X649-labelled platelets from naïve mice (pink) for 1 hour, detached and stained for CD45 (yellow), CD41 (green) and DAPI (blue) before cytospin and imaging. Pink and grey arrows show

surface and internalised platelets, respectively. Data representative of 2 independent experiments with 3 technical replicates and >10 FOV imaged for each condition. **(C)** Naïve or *S. mansoni* infected (10-12 weeks) female C57BL/6 mice were injected with anti-GPIb-V-IX conjugated DyLight 649 (X649) 48hrs prior to harvest and X649$^+$ leukocytes quantified as % of CD45$^+$ cells in blood, liver and spleen. **(D)** Representative flow cytometry of CD41 and X649 expression by classical and non-classical blood monocytes (CD11b$^+$ CD115$^+$ SiglecF$^-$ Ly6G$^-$ Ly6C$^{+/-}$) in naïve or infected mice. Control represents infected mice receiving PBS injection rather than X649. **(E)** Quantification of blood monocytes that are X649$^+$ CD41$^+$ (red) or X649$^+$ CD41$^-$ (grey). **(F)** As E, for liver Ly6C+ (mono) and Ly6C- (Mφ) cells (gated as CD11b$^+$ CD64$^+$ SiglecF$^-$ Ly6G$^-$ Ly6C$^{+/-}$). Data in C-F pooled from 2 experiments with n = 5-8 per group. **(G)** Sorted liver mono-macs from infected mice injected with X649 and stained as **A**. Note combined CD41 and X649 fluorescence appears white on the cell surface whereas X649 alone (internalised platelets) appears pink (see S4E-S4F Fig for individual channels). Data is from single experiment with cells from 2 infected mice and >20 FOV analysed. Significance in C, E and F determined by unpaired *t*-tes*t*. Fig 4A created with BioRender.

Ly6C$^+$ classical monocytes were numerically the most abundant X649$^+$ immune cell in the blood, reflecting their marked expansion following schistosome infection (Fig 4E) [49]. Unlike the spleen, the proportion of liver immune cells that were X649$^+$ increased following schistosome infection (Fig 4C) and in contrast to the blood these were mostly negative for CD41 suggesting platelet internalisation (S4D Fig). The highest proportion of liver X649$^+$ cells were also monocytes/macrophages (Fig 4F, S4D Fig), and whilst these were similar between naive and schistosome infected mice, the substantial increase in total monocytes/macrophages and eosinophils in the schistosome-infected liver ensured the numbers of X649$^+$ cells expanded markedly (Fig 4F, S4D Fig). We sorted CD64$^+$ liver monocytes/macrophages from infected mice and were able to detect those that had internalised X649 + platelets as well as cells with surface-associated platelets, presumably in the process of phagocytosis (Fig 4G, S4E-S4F Fig). Together, these experiments reveal that thrombocytopenia is associated with enhanced platelet uptake by expanded myeloid cell populations, particularly monocyte/macrophages, and that this occurs (at least in part) in the liver.

### 3.5 Platelet-interacting myeloid cells show distinct activation phenotypes

The consequences of platelet-myeloid cell interactions have been extensively studied [16] in acute models of inflammation (predominantly type 1) but less is known about how platelets impact type 2 immune cells [12–14]. To address this, we compared the phenotype of CD41$^-$ and CD41$^+$ cells, focusing on macrophages and eosinophils as key immune cells expanded in schistosome infection. Following *S. mansoni* infection, a greater proportion of liver macrophages and monocytes were positive for surface CD41 (Fig 5A, S5A Fig), consistent with our X649 results (Fig 4). The proportion of MHCII$^+$ and RELMα$^+$ monocytes and macrophages were increased in infected mice (Fig 5B-5C, S5B-S5C Fig) consistent with maturation and alternative activation. Similarly, infection also resulted in a greater proportion of RELMα$^+$ liver eosinophils (S5G Fig). Pairwise analysis of CD41$^+$ vs CD41$^-$ cell types from the same animal revealed substantial differences of these activation markers. CD41$^+$ macrophages (and to a less extent monocytes) from infected mice had lower proportions that were MHCII$^+$ but more that were RELMα$^+$ compared with their CD41$^-$ counterparts (Fig 5D-5E, S5D-S5E Fig). Conversely, whilst CD41$^+$ eosinophils were less prominent than macrophages, a reduced proportion were RELMα$^+$ compared to CD41$^-$ eosinophils (S5F-S5H Fig). To further test for differential activation, we sorted CD41$^-$ and CD41$^+$ monocytes/macrophages from livers of schistosome-infected mice and compared gene expression for a range of type 1 and type 2 inflammatory molecules. This revealed CD41$^+$ liver macrophages/monocytes have an unusual phenotype with increases in both pro- (*Il6*) and anti-inflammatory factors (*Retlna*), but downregulated other facets of pro-inflammatory myeloid cells (i.e., reduced *Tnf*) (Fig 5F). CD41$^+$ macrophages/monocytes also expressed more *Cxcl12* which has been linked to a pro-angiogenic and regulatory phenotype [50]. Whilst this data indicates CD41$^+$ myeloid cells are phenotypically distinct, it does not address whether platelets directly modulate myeloid cell activation [14] or alternatively whether they preferentially bind these cells (due to, e.g., distinct surface receptor expression). To test whether platelets directly modulate macrophage alternative activation, we cultured BMMφ with varying amounts of IL-4 either alone or in the presence of platelets from naive or schistosome infected mice. As expected, IL-4 caused a dose-dependent increase in macrophage RELMα expression but neither set of platelets altered this (S5I Fig). To assess whether schistosome molecules facilitate any impact of platelets on macrophage activation, we treated BMMφ -/+ schistosome egg antigen (SEA) in the presence of IL-4, again with no platelets or platelets from naive or

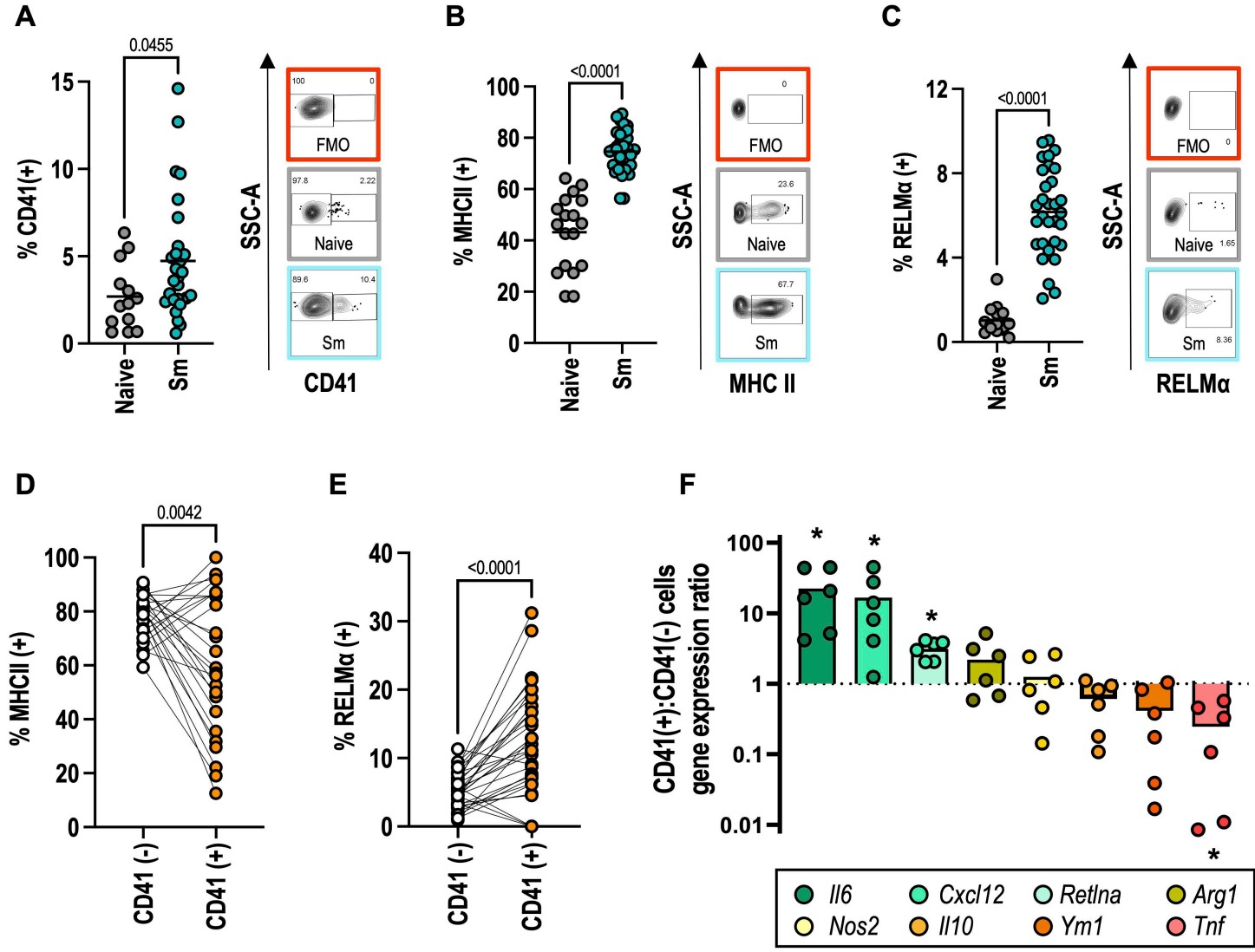

**Fig 5. CD41⁺ myeloid cells have a distinct activation phenotype. (A)** CD41, **(B)** MHC II and **(C)** RELMα expression by liver macrophages (gated CD11b⁺ CD64⁺ SiglecF⁻ Ly6G⁻ Ly6C⁻ cells) from naïve female C57BL/6 mice or following infection with 40-50 *S. mansoni* cercariae (10-12 weeks). Data pooled from 4 experiments (n = 4-9 per group) with representative flow cytometry including fluorescence minus one (FMO) controls shown. Pairwise comparison of **(D)** MHC II and **(E)** RELMα expression by CD41⁻ (white) vs CD41⁺ (orange) liver macrophages from infected mice. Data is pooled from 3 experiments. **(F)** qPCR analysis of indicated transcript expression by sorted CD41⁺ vs CD41⁻ liver mono/macrophages (CD45⁺ CD11b⁺ F4/80⁺ SiglecF⁻ Ly6G⁻ CD41⁻ʹ⁺) shown as CD41⁺:CD41⁻ ratio. Data pooled from 2 independent experiments with n = 3 per group and each data point represents cells from individual animal. Significance in A-C and F determined by unpaired *t*-test, and in D-E by paired *t*-test between individual animals.

infected mice. Surprisingly, SEA contains a factor that significantly reduces BMMφ RELMα expression, and far from increasing RELMα, platelets from schistosome-infected mice caused further (minor) reduction (S5J Fig). Together, these *in vitro* experiments indicate platelets do not drive enhanced macrophage RELMα expression observed during *in vivo* infection.

### 3.6 Experimental manipulation of platelet levels reveals their essential role in schistosome infection

We next wanted to test the *in vivo* immunomodulatory potential of platelets in type 2 immune responses during schistosome infection. We trialled recombinant human TPO treatment between weeks 8 and 9 post-infection but this did not substantially

expand platelet numbers (S6A Fig), although liver myeloid cell PLA were increased (S6B Fig). As an alternative, we treated mice with the TPO mimetic romiplostim once a week between weeks 6–9 post-infection. This caused substantial and sustained increases in platelet numbers and MPV (Fig 6A-6B, S6C Fig). Notably, platelet numbers were higher in naive mice treated with romiplostim than their infected counterparts. Furthermore, we detected marked increases in CD41$^+$ MK in the BM as well as in the spleen and liver (Fig 6C, S6D Fig). Romiplostim signals through the TPO receptor (Mpl) which is expressed by BM progenitors alongside MK and platelets [51]. As such, we next looked at the impact of romiplostim on hematopoietic stem and progenitor cells. Consistent with recent reports [52] we found schistosome infection causes expansion of BM immune cell progenitors (S6E-S6F Fig), and this is further boosted by romiplostim treatment, specifically LT-HSC, MK/erythroid-biased MPP2 and granulocyte/monocyte-biased MPP3 cells (S6G Fig). Given these marked changes to both platelets and immune progenitors, we checked for gross pathological changes in infected mice, but found romiplostim treatment did not alter hepatosplenomegaly or liver granulomas (S6H-S6J Fig). Whilst previous studies indicate platelets facilitate parasite egg excretion [53], romiplostim did not impact intestinal or liver egg counts (S6K Fig), although we did not measure fecal egg excretion. Platelets have been shown to aid immune cell recruitment to inflammatory sites [11,14,54–56] and here we show romiplostim significantly increased liver monocyte and neutrophil PLA, with a strong trend to increase also seen for macrophages (Fig 6D). This was associated with a moderate increase in liver neutrophils but no change in monocytes, macrophages or other immune cells (S6L Fig). Finally, romiplostim did not alter liver monocyte and macrophage MHCII expression, macrophage and eosinophil RELMα (Fig 6E, S6M Fig), or CD4 T cell cytokine production (Fig 6F, S6N Fig). Taken together, this shows that whilst stimulating TPO-receptor signalling boosts platelet numbers, promotes PLA formation and is associated with marked BM changes, this does not alter type 2 immune responses in schistosomiasis.

As a complementary approach, we assessed the impact of platelet depletion using anti-CD41 mAb treatment. As previous studies have shown antibody-mediated platelet depletion causes extensive mortality following high-dose schistosome infection (150–200 cercariae) [53], we reduced our infective dose to ~30 cercariae (compared to standard ~40–50). Despite this precaution, 50% of mice given anti-CD41 mAb (3/6) developed symptoms of severe disease (weight loss, hunched appearance, pallor) and were culled (Fig 7A). As an alternative, we infected Mpl$^{-/-}$ mice which lack TPO receptor signalling and so are severely thrombocytopenic [41,57–59]. We again used a low dose of cercariae and harvested at 7.5 weeks post-infection due to concerns about the development of severe disease. Infected WT mice showed little change in platelet numbers at this early timepoint after lower dose infection although MPV was still increased, particularly in Mpl$^{-/-}$ animals (Fig 7B-7C). In addition, infected Mpl$^{-/-}$ mice developed anaemia and elevated red cell distribution width (S7A-S7B Fig), coincident infection-induced intestinal bleeding (S7C Fig). This bleeding pathology in Mpl$^{-/-}$ was not associated with enhanced hepatosplenomegaly (S7D-S7E Fig), liver egg burden (S7F Fig) or granulomatous inflammation (S7G Fig). Instead, infected Mpl$^{-/-}$ mice showed severe reductions in BM LSK immune progenitors (S7H-S7I Fig), which is consistent with their increase in romiplostim-treated mice BM (S6E-S6F Fig). Given this widespread hematopoietic dysfunction it was somewhat surprising then that there was no significant difference in the number of mature immune cells in the liver of infected WT and Mpl$^{-/-}$ mice (S7J Fig). Importantly, liver myeloid cells (particularly monocytes and macrophages) from Mpl$^{-/-}$ mice showed reduced (but not absent) CD41 positivity consistent with reduced PLA formation (Fig 7D and S7K Fig). We measured myeloid cell activation markers and found a lower proportion of MHCII$^+$ monocytes and macrophages in Mpl$^{-/-}$ mice following schistosome infection (Fig 7E). Whilst there was no significant difference in the proportion of RELMα$^+$ macrophages, there was an increasing trend (p = 0.07) in Mpl$^{-/-}$ eosinophils following infection (S7L Fig). We confirmed there was no difference in *Retnla* expression (encoding RELMα) between WT and MPL$^{-/-}$ liver macrophages using qPCR and also, in marked contrast to the phenotype of CD41$^+$ liver macrophages (Fig 5F), found MPL$^{-/-}$ macrophages express lower levels of *Tnf* and elevated *Il6* compared with their WT counterparts (S7M Fig). Finally, CD4 T cell responses were elevated in infected Mpl$^{-/-}$ with significant increases in IFNγ and trends for more IL-4 and IL-10, despite reduced MHCII expression (Fig 7F and S7N Fig). Together this shows that genetic impairment of platelet production reduces myeloid cell PLA and alters maturation marker expression.

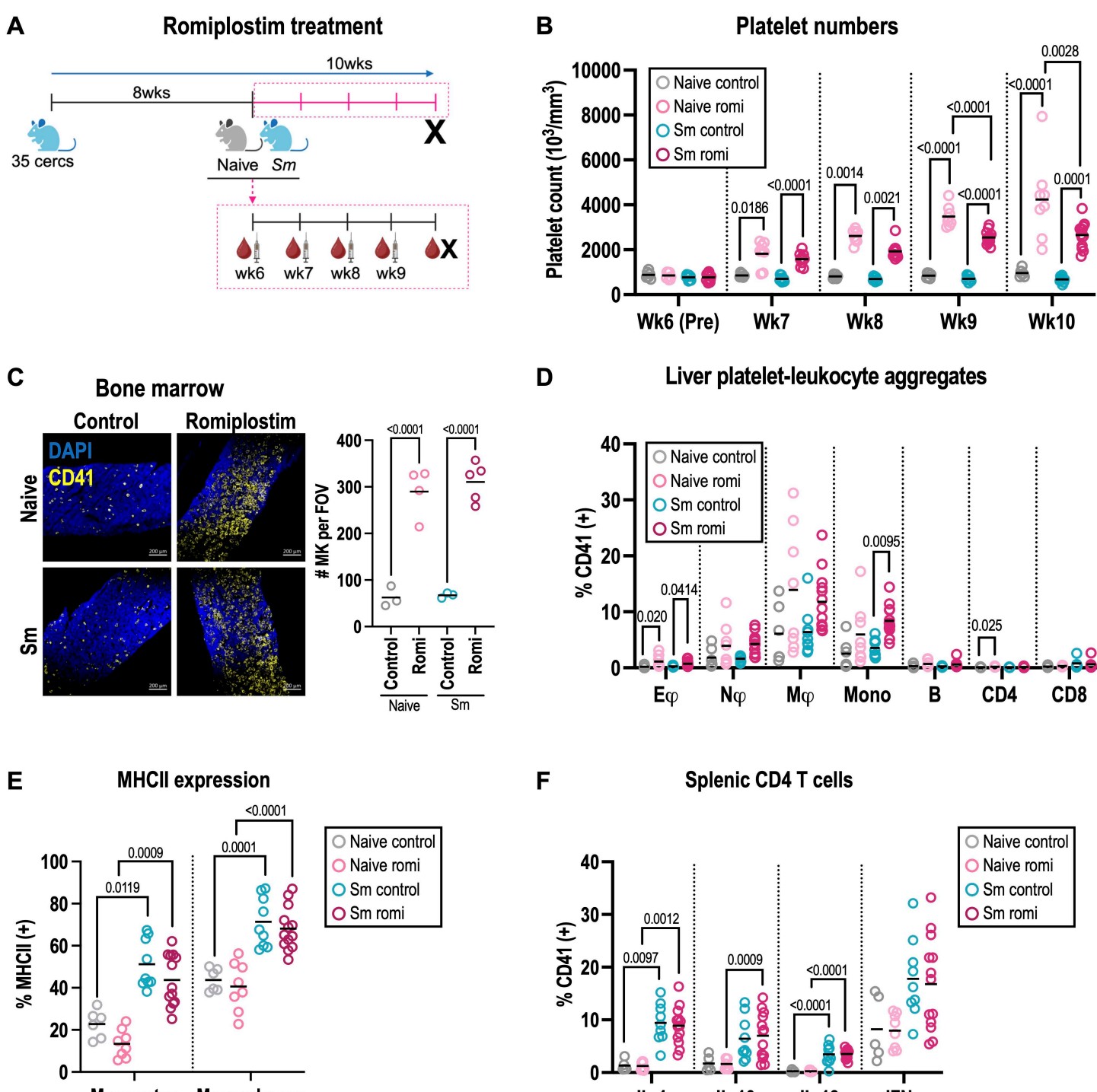

**Fig 6. MPL signalling promotes platelet production but does not alter type 2 immunity. (A)** Female C57BL/6 mice -/+ *S. mansoni* were treated with romiplostim or control saline at weeks 6, 7, 8 and 9 post-infection. Blood samples were taken before and 4 days after each treatment. Mice were harvested at wk10 post-infection. Naive controls are grey, naive romiplostim pink, infected control blue and infected romiplostim red **(B)** Longitudinal blood platelet counts. **(C)** Representative BM sections stained for CD41 (yellow, MK) and DAPI (blue, nuclei) at wk10 with MK quantification per field of view (FOV). **(D)** % of indicated liver immune cells that are CD41+ (gating as S4 Fig). **(E)** MHC II expression by liver monocytes (Ly6C+) and macrophages (Ly6C-). **(F)** Production of indicated intracellular cytokines by splenic CD4 T cells following *ex vivo* stimulation. Data pooled or representative of 2 experiments with n = 4-7 per group, significance in B-F determined by ANOVA with Tukey post-hoc test. Fig 6A created with BioRender.

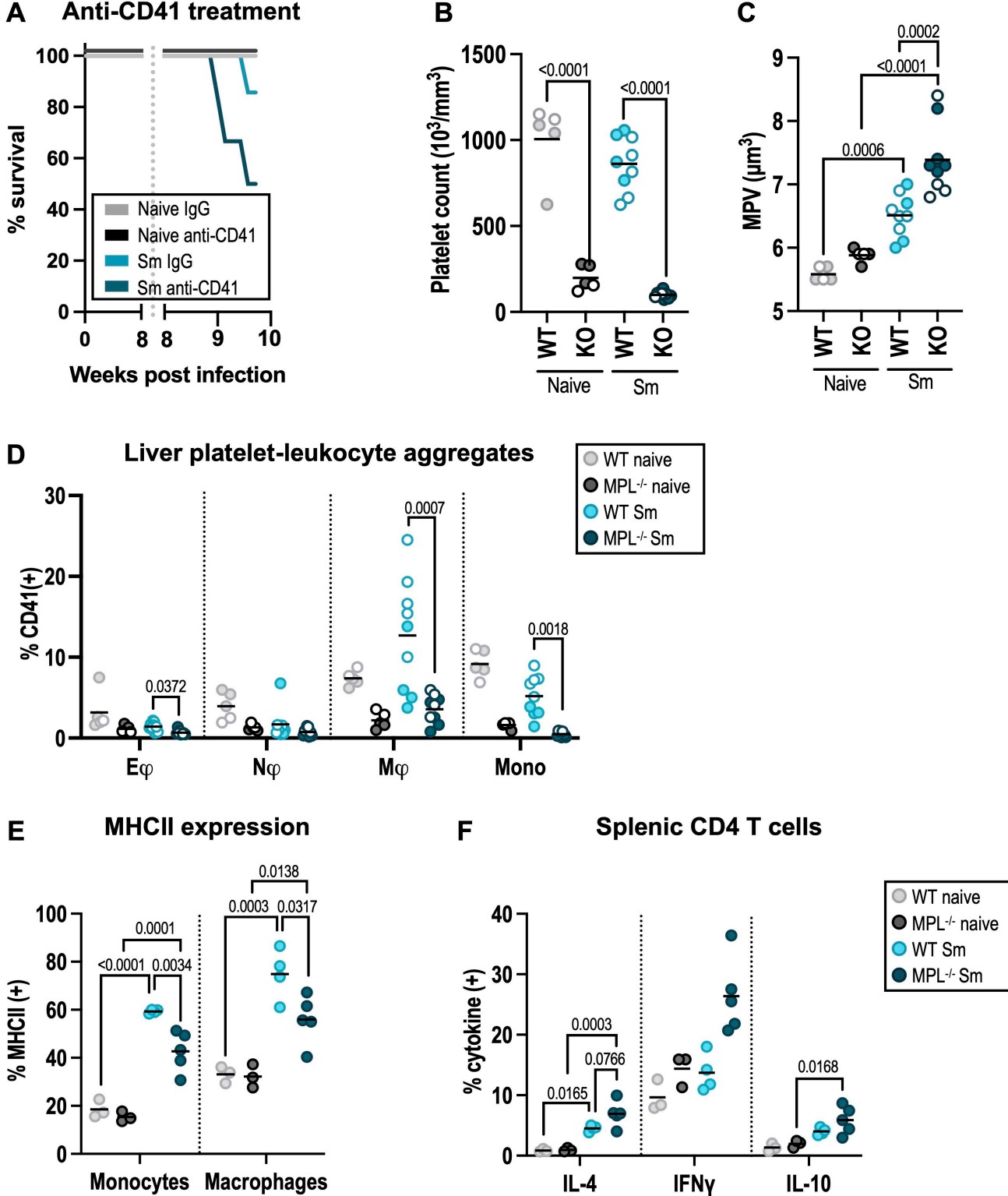

**Fig 7. Platelet depletion impacts myeloid cell phenotype and host survival. (A)** Survival of naive and *S. mansoni* infected WT mice treated with a depleting anti-CD41 mAb or rat IgG isotype control from wk8 post-infection. **(B-F)** WT or Mpl⁻/⁻ mice -/+ infection with 30 *S. mansoni* cercariae for 7.5 weeks (open circles males, closed circles females). **(B)** Platelet counts and **(C)** MPV determined at harvest. **(D)** % of liver eosinophils (Eφ), neutrophils

(Nφ), macrophages (Mφ) and monocytes (mono) that are CD41[+]. Gating as S4 Fig (E) Expression of MHC II by liver monocytes and macrophages. (F) Production of indicated intracellular cytokines by splenic CD4 T cells following *ex vivo* stimulation. Data in (D-F) pooled or from 2 mixed sex experiments with n = 2-3 (naives) or 4-5 (infected) per group (open symbols male, closed symbols female). Data in (E-F) from female experiment. Significance determined by ANOVA with Tukey post-hoc test.

## 4. Discussion

Platelets have key roles in the initiation and regulation of inflammatory immune responses following bacterial and fungal infections, as well as in allergic responses [12,14–16,56]. However, whether platelets modulate type 2 immune responses to helminth parasites is less well understood. Using a mouse model of chronic schistosomiasis, we find that infection promotes the formation of platelet-leukocyte (macrophages/monocytes) aggregates, and this is associated with platelet clearance and thrombocytopenia. Whilst platelets preferentially interact with liver macrophages with a distinctive activation status (i.e., enriched in RELMα[+] MHC II[+] cells), *in vitro* and *in vivo* studies indicate platelets have only limited impact on macrophage phenotype.

Schistosome infection is characterised by type 2 immune responses with elevated IL-4, IL-5, IL-13 and IgE [60]. Similar responses occur in allergic inflammation, where platelets have important roles in the recruitment of inflammatory cells (particularly eosinophils) as well as the initiation of adaptive immunity [35,56]. We find that schistosome infection causes thrombocytopenia and elevated MPV, which may reflect BM stress [61] and/or the preferential clearance of smaller mature platelets. Whilst infection does not alter BM MK cell numbers or *in vitro* MK lineage bias, it does reduce circulating TPO, lead to smaller MK size, and a bias towards lower ploidy cells with increased LSP-1[+] expression. Such MK have been previously shown to have lower platelet-producing capacity and elevated inflammatory immune function [43], but further studies are required to determine whether this is the case here. TPO is well known to drive MK-biased HSC development [62] but alternative pathways exist with IL-6 and IL1β also capable of promoting megakaryopoiesis and platelet production [63]. The impact of reduced TPO levels on other *Mpl*-expressing cell populations that include large numbers of hematopoietic stem and progenitor cells [64] is as yet unknown. In this regard, schistosome infection has been recently shown to strongly activate other BM cells including HSC and downstream mature progenitors [65]. Similarly, excretory/secretory molecules from the trematode *Fasciola hepatica* promotes innate immune training of BM HSCs that lead to more muted inflammatory responses [66]. Importantly, infection-induced changes to TPO and platelets do not revert after PZQ treatment, suggesting the impact of schistosomiasis on hematopoiesis is stable for several weeks after parasite death, likely due in part to persistent disease pathology (e.g., liver fibrosis). Extended time course experiments will be required to assess the durability of these changes.

We next assessed whether accelerated platelet clearance contributed to platelet depletion in infection. Using an *in vivo* platelet tracking system [67] we demonstrated platelets are cleared more rapidly following schistosome infection, and this was even more dramatic in visceral leishmaniasis. Despite schistosomiasis leading to high levels of antibody production [68] and previous studies [27] demonstrating antigenic cross-reactivity between parasite eggs and platelets, studies in FcRγ[-/-] mice show this is not mediated by activating Fc receptors (FcγRI/III/IV, FcεRI and FcαR1). Our *in vivo* platelet tracking also helped identify a notable increase in PLA during infection, particularly with classical monocytes in the blood, suggesting enhanced immune cell interactions may contribute to reduced platelet levels. Co-staining for surface CD41 allowed us to identify whether platelets were surface-bound or internalised, and this showed infection promoted platelet uptake by liver monocytes and macrophages. Whilst this implicates the liver and blood as sites of platelet clearance and/or immune cell binding, further experiments with splenectomised mice would be require to test any role for the spleen in this process. PLA have been reported in pulmonary inflammation and cardiovascular disease, but surprisingly the eosinophil-platelet interactions that occur in allergy were relatively rare [14,54,69,70].

We found platelet-associated macrophages to be enriched for a RELMα[+] and MHC II[-] phenotype, yet *in vitro* co-culture experiments indicated platelets have minimal effects on macrophage alternative activation. Additionally, platelets from infected mice actually slightly reduced macrophage RELMα levels in the presence of schistosome egg antigen. Moreover, whilst

platelet-interacting liver macrophages in WT mice express elevated *Il6* and reduced *Tnf*, this is also the case for macrophages from platelet-deficient Mpl⁻/⁻ animals. This discrepancy likely reflects TPO signalling having multiple roles in hemostasis and immune cell development, but also shows that the presence or absence of platelets is not the primary determinant of macrophage phenotype. Our findings contrast with human studies where platelets convert monocytes to an M2-like phenotype [71], although notably human and mouse markers of alternate activation differ markedly. Whilst future transcriptomic studies may reveal global gene expression signatures associated with platelet-binding expanding those reported here, it will be important to separate cause (i.e., platelet-mediated immunomodulation) from consequence (i.e., differential macrophage receptor expression promoting platelet binding). To answer this, we assessed immunological and parasitological parameters after *in vivo* manipulations of platelet levels. Treatment of mice with romiplostim substantially increased platelet numbers, although this was somewhat lower in infected mice likely as a result of accelerated platelet clearance, immune cell interactions, and potentially infection-induced changes to MK. Importantly, romiplostim did not alter macrophage phenotype (as judged by MHCII and RELMα expression), inflammatory immune cell recruitment to the liver or parasitological parameters (liver pathology, parasite egg production). This is striking because romiplostim strongly promoted BM hematopoiesis with marked expansion in HSC and more mature progenitors. Others have shown that romiplostim negatively impacts long-term HSC output [62], consistent with HSC attrition when driven into cell cycle [72]. We also performed complementary experiments to deplete platelets during infection. Using a well-characterised anti-CD41 mAb mediated immune thrombocytopenia (ITP) model, we found platelet-depletion substantially reduced host survival in schistosome infection, consistent with earlier studies using anti-platelet polysera [53]. As an alternative, we turned to Mpl⁻/⁻ mice that have intrinsically low platelet counts [63,73]. Again, we observed substantial ill health following infection (anaemia, intestinal hemorrhaging) that led us to harvest at an early time-point (7.5 weeks). This can develop in chronic infection (i.e., >10wks) or following exposure to high number of parasites (100–200 cercariae) and suggests Mpl⁻/⁻ mice have insufficient platelets to stem intestinal bleeding caused by transit of schistosome eggs. We also observed a complete failure of BM progenitor cell populations to respond to infection together with elevated CD4⁺ T cell IFNγ production. We hypothesise the increased type 1 immune response in Mpl⁻/⁻ mice reflects enhanced gut bacterial translocation due to intestinal bleeding, although further studies are required to test this. In this regard, elevated circulating LPS levels have been noted in both mouse [74] and human schistosomiasis [75], although this does not occur in less severe disease [76].

Separating hemostatic and immunological roles for platelets in schistosomiasis is complicated. Platelet depletion interferes with hemostasis essential to survive infection, alternatives such as platelet-specific cre-drivers have off-target effects on other cells [77,78], and TPO receptor agonists also impact immune progenitors (as here). Moving forward, our recent proteomic comparison of platelets in naive and schistosome-infected mice provides a set of candidates that may modulate immune cell function, including MHC-I, several complement components, antibodies and other immune-related molecules (e.g., galectin-9, CD84) [79]. Together, our study has revealed schistosome infection accelerates platelet clearance and promotes immune cell interactions. Whilst platelets do not appear to globally regulate type 2 immunity, infection-induced platelet-hepatic monocyte/macrophage interactions provide an exciting focus for future work.

## Supporting information

**S1 Fig. PZQ does not fully reverse schistosome infection-induced changes.** C57BL/6 mice were infected with 40–50 *S. mansoni* cercariae and treated -/+PZQ at week 12 post-infection and 4 weeks later (A) Adult worms counts, (B) Liver and (C) Spleen weight at week 16 post-infection. (D-E) Pairwise comparison of (D) Platelet count and (E) MPV in individual animals at weeks 12 (pre-PZQ) and 16 (post-PZQ) post-infection in naïve, infected (Sm) and infected then PZQ-treated mice. (F) Correlation of platelet count and MPV in naïve, infected (Sm) and infected-cured (PZQ mice) at wk16. (G-H) Day 21 platelet number and MPV in mice injected with schistosome eggs on d0 (intraperitoneal) and d14 (intravenous). Data in B-H is pooled from 2-3 experiments with n = 2–6 per group. Significance determined by unpaired *t*-test (A, G-H), ANOVA with Tukey post-hoc test (B-C), paired *t*-test (D-E) or linear regression (F).
(TIFF)

**S2 Fig. IL-4, but not schistosome infection, impairs megakaryopoiesis.** C57BL/6 mice were infected or not with *S. mansoni* (*Sm*) for 10 weeks or *L. donovani* (*Ld*) for 4 weeks and (A) platelet counts, (B) qPCR liver *Thpo* expression (C) BM MK diameter and (D) BM CD68$^+$ macrophage cell number per field of view (FOV) determined. (E-F) Bone marrow MK expression of CD53, LSP-1, ARNTL and MYLK4 in naïve and schistosome infected mice determined by flow cytometry. FMO control stains are shown in light grey. (G) Representative colony formation assays using BM HSC (Lin$^-$ c-Kit$^+$ EPCR$^+$ Sca1$^+$ CD48$^-$ CD150$^+$) from naive and 12 week schistosome-infected mice -/+ exogenous IL-4. (H) Total colony count and (I) CD41$^+$ cells within each colony are shown. (J) Platelet counts and (K) MPV in mice treated with IL-4c or PBS control as detailed in Materials and Methods. Data are pooled (A-D, H), representative (G, I) of 2–3 experiments, or from a single experiment (J-K). Significance is determined by ANOVA with Tukey post-hoc test, Kruskal-Wallis with Dunn's test or unpaired t-test.
(TIFF)

**S3 Fig. Schistosome-induced thrombocytopenia does not require activating Fc receptors.** (A) Representative flow cytometry histograms showing X649 labelling of gated CD41$^+$ platelets for naïve (black), 12 wk *S. mansoni* (blue) and 4wk *L. donovani* (orange)-infected mice at 1, 24 and 48 hours post-X649 treatment. (B) Platelet counts in WT and FcRγ$^{-/-}$ mice that are either naïve or 12 wk *S. mansoni* infection. (C) Platelet counts in WT and FcRγ$^{-/-}$ mice pre and 24 hours post-treatment with isotype control (-) or a depleting anti-CD41 mAb (+). Data representative of 2–3 experiments (A-B) or are a single experiment (C). Significance is determined by ANOVA with Tukey post-hoc test (B) or paired *t*-test comparing individual mice (C).
(TIFF)

**S4 Fig. Identification of platelet-interacting immune cells.** (A) BM-Mφ were co-cultured with X649-labelled platelets from naïve mice. Macrophages were recovered 0.5, 1, 2, 4, 8 and 24 hours later and stained for CD11b, CD64 and CD41. Proportion of macrophages (CD11b$^+$ CD64$^+$) that are X649$^+$ shown in pink (left axis) and proportion of X649$^+$ macrophages that have CD41$^+$ surface stain shown in green (right axis). (B) Representative flow cytometry of macrophages following X649$^+$ platelet co-culture for 0.5 and 24 hours with X649$^+$ (pink) and surface CD41$^+$ (green) indicated. Data in A-B from single experiment with 6 technical replicates at every time point, bars represents mean +/- SEM. (C) Naïve or *S. mansoni* infected (10–12 weeks) C57BL/6 mice were injected with anti-GPIb-V-IX conjugated DyLight 649 (X649) 48hrs prior to harvest. Blood X649$^+$ leukocytes that are either CD41+ (pink) or CD41- (grey) in the blood are quantified and shown as proportion of each cell population. Monocytes are gated as CD11b$^+$ CD115$^+$ SiglecF$^-$ Ly6G$^-$ Ly6C$^{+/-}$, eosinophils are CD11b$^+$ SiglecF$^+$ SS$^{hi}$, neutrophils as CD11b$^+$ Ly6G$^+$ SiglecF$^-$ SS$^{hi}$, B cells as CD19$^+$ TCRb$^-$, CD8 T cells as TCRb$^+$ CD8a$^+$, and CD4 T cells as TCRb$^+$ CD4$^+$. (D) As C for liver immune cells showing absolute number of different immune cells per liver, % of each cell type that is X649$^+$, and number of each cell type that is X649$^+$. Individual channels from Fig. 4G showing E) cell surface and F) internalised platelets associated with macrophage. Data in C-D pooled from 3 experiments and significance determined by unpaired *t*-test with comparisons between stacked bars colour coded.
(TIFF)

**S5 Fig. Whilst platelet-associated myeloid cells have altered activation phenotypes, platelets do not promote macrophage RELMα expression.** (A) CD41, (B) MHC II and (C) RELMα expression by liver monocytes (gated CD11b$^+$ CD64$^+$ SiglecF$^-$ Ly6G$^-$ Ly6C$^+$ cells) from naïve C57BL/6 mice or following infection with *S. mansoni* (10–12 weeks). Data pooled from 4 experiments. Pairwise comparison of (D) MHC II and (E) RELMα expression by CD41$^-$ (white) vs CD41$^+$ (orange) liver monocytes from infected mice with data pooled from 3 experiments. (F) CD41 and (G) RELMα expression by liver eosinophils (gated CD11b$^+$ SiglecF$^+$ SS$^{hi}$) as A-C. (H) Pairwise comparison of RELMα expression by CD41$^-$ (white) vs CD41$^+$ (orange) eosinophils from infected mice. (I) BMMφ were cultured with indicated concentrations of IL-4, either alone (white) or with platelets from naïve (grey) mice or mice infected for 12wks with *S. mansoni* and RELMα expression

determined 18hrs later by flow cytometry. Data pooled from 2 experiments and expression is normalised to 10ng/ml IL-4 (no platelets) group. (J) BMMφ were cultured with (grey) or without (white) SEA for 18hrs in the presence of 10ng/ml IL-4, either in the absence of platelets or with platelets from naïve or schistosome-infected mice. Significance in A-C and F-G determined by unpaired *t*-test, (D-E, H) by paired *t*-test between individual animals, and (J) by two-way ANOVA (*=p<0.001 vs no SEA, †=p<0.05 vs no platelets).
(TIFF)

**S6 Fig: MPL signalling boosts HSPC subsets but does not alter infection parameters.** C57BL/6 mice -/+ *S. mansoni* were treated with human TPO 3x per week at week 8 and 9 post-infection with (A) blood platelets and (B) liver CD41+ platelet-leukocyte aggregates (PLA) determined at wk10. (C) Platelet MPV in mice treated with romiplostim as Fig. 6A. (D) Representative liver (left) and spleen (right) sections to identify MK (nucleated CD41+ cells) with MK quantified per field of view (FOV). Livers stained for CD41 (yellow), TPO (red, hepatocytes), DAPI (blue, nuclei). Spleens stained for CD41 (yellow), B220 (green, B cells), F4/80 (red, macrophages), DAPI (blue, nuclei). (E) Representative flow cytometry BM LSK cells (lineage- Sca1+c-kit+) and (F) % LSK of live BM cells in control (ctrl) and romiplostim (R) treated naive and infected (Sm) mice. (G) BM LSK subsets (LT-HSC=CD135- CD150+ CD48-; ST-HSC=CD135- CD150- CD48-; MPP2=CD135- CD150+ CD48+; MPP3=CD135- CD150- CD48+; MPP4=CD135+ CD150- CD48+). (H-I) Liver and spleen weights in control and romiplostim-treated mice -/+ *S. mansoni* infection. (J) Quantification of granuloma coverage (left) and granuloma diameter (right) in livers from control or romiplostim-treated mice. (K) liver (left) and intestine (right) parasite eggs. (L) Number of the liver CD41+ leukocytes. (M) Percentage RELMα+ macrophages and eosinophils. (N) Percentage of liver CD4 T cells that are positive for the indicated intracellular cytokine following ex vivo stimulation. Data pooled or representative of 2 experiments, significance determined by unpaired t-test (J-K) or ANOVA with Tukey post-hoc test (other panels).
(TIFF)

**S7 Fig. Mpl deficiency impacts hematopoiesis and host survival.** (A) Red blood cell count and (B) Red Density Width (RDW) in naive and *S. mansoni*-infected WT and Mpl-/- mice. Mice were infected with a lower dose on cercariae (30) and harvest at an earlier point (7.5wks) than other experiments to avoid host death. (C) Intestinal bleeding (arrow) in infected Mpl-/- mice, whereas liver pathology was seen in both WT and Mpl-/-. (D) Liver and (E) spleen weight in naïve and *S. mansoni*-infected WT and Mpl-/- mice. (F) Liver eggs (G) liver granuloma coverage and size of single egg granulomas in WT and Mpl-/- mice. (H) Representative flow cytometry of BM LSK cells (lineage- Sca1+c-kit+) and (I) numbers of BM LSK in naive and infected WT and Mpl-/- mice. (J) Numbers of liver eosinophils (Eφ), neutrophils (Nφ), macrophages (Mφ) and monocytes (mono). (K) Representative CD41 flow cytometry staining for liver macrophages. (L) Liver macrophage and eosinophil RELMα determined by flow cytometry. (M) Z-scores of indicated transcripts in liver macrophages from infected WT and Mpl-/- mice determined by qPCR (N) Representative spleen CD4 T cells intracellular cytokine staining. Data for (A-F, H-L) pooled from two experiments and between male and female mice (open symbols male, closed symbols female). Significance determined by ANOVA with Tukey post-hoc test or unpaired *t*-test.
(TIFF)

**S1 Table. Extracellular flow cytometry antibodies.**
(TIFF)

**S2 Table. Intracellular flow cytometry antibodies and conjugates.**
(TIFF)

**S3 Table. Primary and secondary antibodies used for confocal microscopy.**
(TIFF)

**S4 Table. Primers and Taqman probes used for qPCR.**
(TIFF)

**S1 Data. Graphpad Prism file containing raw data.**
(ZIP)

## Acknowledgments

We thank staff in the University of York Biological Services Facility (animal husbandry) and Bioscience Technology Facility Imaging & Cytometry laboratory (flow cytometry and confocal microscopy). We also thank Dr. Benjamin Hulme and Prof. Karl Hoffmann (Aberystwyth University) for providing schistosome infected snails.

## Author contributions

**Conceptualization:** Joanna H Greenman, Ian S Hitchcock, James P Hewitson.

**Data curation:** Joanna H Greenman, James P Hewitson.

**Formal analysis:** Joanna H Greenman, Shinjini Chakraborty, Lucie Moss, James P Hewitson.

**Funding acquisition:** Ian S Hitchcock, James P Hewitson.

**Investigation:** Joanna H Greenman, Shinjini Chakraborty, Lucie Moss, James P Hewitson.

**Methodology:** Joanna H Greenman, Shinjini Chakraborty, Lucie Moss, Paul C Armstrong, Paul M Kaye, James P Hewitson.

**Supervision:** Paul M Kaye, Ian S Hitchcock, James P Hewitson.

**Visualization:** Joanna H Greenman, Shinjini Chakraborty, James P Hewitson.

**Writing – original draft:** Joanna H Greenman, Ian S Hitchcock, James P Hewitson.

**Writing – review & editing:** Joanna H Greenman, Shinjini Chakraborty, Paul M Kaye, Ian S Hitchcock, James P Hewitson.

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
