## [Decision Letter · Decision Letter 0]

16 Jan 2025

Thrombocytopenia in murine schistosomiasis is associated with platelet uptake by liver macrophages that have a distinct activation phenotype

Dear Dr. Hewitson,

Thank you for submitting your manuscript to PLOS Pathogens. After careful consideration, we feel that it has merit but does not fully meet PLOS Pathogens's publication criteria as it currently stands. Therefore, we invite you to submit a revised version of the manuscript that addresses the points raised during the review process.

Please submit your revised manuscript within 60 days Mar 17 2025 11:59PM. If you will need more time than this to complete your revisions, please reply to this message or contact the journal office at plospathogens@plos.org. Please include the following items when submitting your revised manuscript:

We look forward to receiving your revised manuscript.

Kind regards,

Constance A. M. Finney

Academic Editor

PLOS Pathogens

James Collins III

Section Editor

PLOS Pathogens

Editor-in-Chief

PLOS Pathogens

orcid.org/0000-0003-2946-9497

Editor-in-Chief

PLOS Pathogens

orcid.org/0000-0002-7699-2064

**Journal Requirements:**

At this stage, the following Authors/Authors require contributions: Joanna H Greenman, Shinjini Chakraborty, Lucie Moss, Paul C Armstrong, Paul M Kaye, Ian S Hitchcock, and James P Hewitson. Please ensure that the full contributions of each author are acknowledged in the "Add/Edit/Remove Authors" section of our submission form.

- ® on page: 8.

- TM on pages: 6, 10, and 11.

5) We notice that your supplementary Figures, and Tables are included in the manuscript file. Please remove them and upload them with the file type 'Supporting Information'. Please ensure that each Supporting Information file has a legend listed in the manuscript after the references list.

Potential Copyright Issues:

- Please confirm that you are the photographer of Figure 7C, or provide written permission from the photographer to publish the photo(s) under our CC BY 4.0 license.

- Figues 3A and 6A; Please confirm whether you drew the images / clip-art within the figure panels by hand. If you did not draw the images, please provide (a) a link to the source of the images or icons and their license / terms of use; or (b) written permission from the copyright holder to publish the images or icons under our CC BY 4.0 license. Alternatively, you may replace the images with open source alternatives. See these open source resources you may use to replace images / clip-art:

7) We note that your Data Availability Statement is currently as follows: "All data available on request.". Please confirm at this time whether or not your submission contains all raw data required to replicate the results of your study. Authors must share the “minimal data set” for their submission. PLOS defines the minimal data set to consist of the data required to replicate all study findings reported in the article, as well as related metadata and methods (https://journals.plos.org/plosone/s/data-availability#loc-minimal-data-set-definition).

- The points extracted from images for analysis..

**Reviewers' Comments:**

Reviewer's Responses to Questions

**Part I - Summary**

Reviewer #1: In their manuscript “Thrombocytopenia in murine schistosomiasis is associated with platelet uptake by liver macrophages that have a distinct activation phenotype”, Greenman and colleagues investigate the role of platelets on macrophage phenotype in schistosome infections. The study provides insights into the role of platelet in macrophage alterations following schistosoma infection. The study is potentitally interesting, however, there are some points that need to be clarified.

Reviewer #2: This is an interesting and novel work which has investigated thrombocytopenia in schistosomiasis. The authors have used an impressive variety of murine intervention models to assess the role of platelets and their interaction with the immune system in schistosomiasis. The work is clearly written, the figures are clear and the science seems well-executed.

Key findings are:

- Schistosomiasis causes thrombocytopenia which is maintained after PZQ treatment

- The causes of the thrombocytopenia are investigated: TPO is reduced in schistosomiasis, although MKs actually increase. This seeming contradiction may be explained by a change in ploidy of the MKs (to a less platelet-producing form). In addition, platelet clearance rate is increased. The increased rate of platelet clearance may be explained by increased binding and internalization of platelets by monocytes and macrophages.

- Platelet-bound macrophages had a different phenotype, although cause and effect is not conclusively determined.

- Increasing platelets (with romiplostin) does not seem to affect immune responses. Decreasing platelets (with anti-CD41 or MPL-/- mice) increases morbidity but also alters immune responses.

A key weakness of the study is that the authors have shown that macrophages associated with platelets have a different phenotype, but it is not clear if this can be attributed to the platelets. E.g. does binding platelets change the macrophage phenotype, or does having a different phenotype allow them to better bind platelets. However I am satisfied that the authors have discussed the difficulties of unpicking this in the final paragraph of the discussion.

Reviewer #3: Overall this is an intriguing study that demonstrates that schistosome infection modulates the lifecycle of platelets and that platelet myeloid interactions occurs during infection. This data fits well with recent work in the field that has demonstrated that schistosome infection modulates different facets of hematopoiesis. While the data presented is of a high quality, there are some lapses and missed opportunities in experimental design that make some of the conclusions in the discussion not well supported. Below are suggestions for data to add that would better support the conclusions around the potential role of platelets in modulating the immune response. The discussion should also better incorporate the entirety of work about parasite infections that modulate different facets of hematopoiesis as there are multiple papers published since 2021 that demonstrate alterations to various lineages.

**Part II – Major Issues: Key Experiments Required for Acceptance**

Reviewer #1: 1) The authors observe an increase in platelet size measured as MPV upon infection. I recommend to confirm that these platelets with increased size are young platelets by Thiazole staining, checking glycoprotein expression and also MHC-I expression

2) Could the authors provide unstained control sections for the red TPO staining in Figure 2A as the liver tissue is highly autofluorescent. Did the authors also quantify TPO levels in liver lysate?

3) Since there is an increased number of MKs in the liver following infection: Do you also see MK precursors and HSCs in the liver?

4) The authors want to study inflammatory megakaryocytes by measuring the ploidy. I recommend using one or more of the markers suggested in the original paper by Sun et al, e.g. CD53

5) What is the authors conclusion where the increased platelet clearance takes place? To prove/exclude the role of liver and spleen I suggest performing cryosections on these tissue and stain for macrophages and platelets to see how many platelets are retained in the respective organs.

6) To investigate altered macrophage phenotype following infection and treatment with romiplostim or in Mpl-deficient animals the authors provide MHCII expression levels. I recommend also checking for other markers shown in figure 5 under these conditions to get a better understanding if there are changes in the macrophage phenotype.

7) Lastly, does romiplostim or Mpl knockout have an impact on the infection/inflammation itself?

Reviewer #2: (No Response)

Reviewer #3: PZQ treatment doesn’t eliminate eggs within the period they examined, due to the time need to resolve granulomas. Remaining eggs in liver and or circulating SEA should be quantified as this phenomenon may be driven specifically by SEA. This could also be addressed via SEA or egg injection of mice. Does that recapitulate the platelet effects or is infection needed?

The IL-4 data is interesting, IL-4c injection for a few weeks would be more relevant in determining its role in the current phenotype than IL-4 exposure in culture

Figure 5 the gene expression profile is interesting, but given the critical role in schistosome pathogenesis for IL-10 it would be easier to place the data in the context of the literature if IL-10 production was demonstrated. The data should at least show IL-10 relative transcript levels for each population (CD41+/-) separately within infected, uninfected control and leishmania infected individuals if IL-10 protein can’t be measured. More minor point, IL-4 driven alternative activation of macrophages is not the same as schistosome driven (different levels of key transcripts like Relm�, Arg1,etc) So the use of IL-4 to activate the macs in the culture experiment does not rule out an effect of the plate interaction on schistosome activated BMDM. Better experiment would be 24 hr SEA stimulation of 7 day BMDM with and without platelets from infected and naive mice

**Part III – Minor Issues: Editorial and Data Presentation Modifications**

Reviewer #1: Minor point: The figure legend to figure 2 says that TPO staining is in yellow, but it is in red.

Reviewer #2: More critical minor issues (e.g. study conclusions should be slightly revised if further analysis or experimentation are not performed):

1. It is stated that TPO production by hepatocytes is impaired, however this is not directly measured. The liver increases in size during schistosomiasis, and so although the area of TPO-producing cells decreases in Fig.2A this does not necessarily reflect a reduction in total TPO producing cells. The TPO brightness in the non-granulomatous liver regions seem brighter after Sm infection (fewer hepatocytes but more TPO?). Better quantification of total TPO production by hepatocytes is required to state that TPO production by hepatocytes is impaired. If this is not done the statement that TPO production by hepatocytes is reduced should be revised.

2. Fig. S5I - no effect observed when platelets where added to BMac. This experiment was designed to determine whether platelets directly modulate macrophage alternative activation. The experiment seems sorely lacking a positive control. Specifically the authors state that in other conditions platelets directly impact macrophage activation - could you apply these conditions to the platelets before adding to the BMacs? I worry that the BMacs do not reflect liver macrophages, with the platelets perhaps not interacting at all with the BMacs (in contradiction to your in vivo findings!). Without a positive control I do not believe you can use this experiment to make any statement about whether platelets do or do not impact macrophage activation.

Actually minor issues:

1. Please add references for the mouse strains used (p6, line 111-113) to allow readers to get further information on the strain and genetics if required.

2. Page 8 line 148 it states control mice received vehicle alone but the vehicle is not stated. What is the vehicle?

3. Page 14 line 290 - Figure S1B is referenced after the statement "TPO is produced primarily by hepatocytes in the liver, which is the major site of granulomatous inflammation in S. mansoni infection". Figure S1B is liver weights and so does not show TPO production and is not (save incredibly indirectly) a measure of granulomatous inflammation. Please use literature references (or some of your own more-direct data) instead.

4. Figure 3C - what do the significance stars refer to? Are you comparing between groups or between timepoints within groups - please clarify.

5. Figure 4D - what is the control column? X649 not added? Please clarify.

6. Line 440 page 20 - is Retlna an anti-inflammatory factor? Elsewhere in the paper you refer to it as an alternative (M2) or Th2 associated factor which seems more accurate to me.

7. Line 633 page 27 - the hypothesis that intestinal bleeding allows gut bacterial translocation and therefore an elevated type 1 response comes a bit out of the blue. Can you provide more evidence either from literature or your own research to support this.

8. Similarly the mentions of pseudo-thrombocytopenia seem to come a bit out of the blue. It would be helpful to include a definition of what you mean by this term. My understanding is that it is a clinical term to refer to spurious/misdiagnosis of thrombocytopenia due to clumping of platelets either alone or with leukocytes. What you are seeing seems to be a real biological phenomenon of elevated myeloid numbers in Sm (in blood but also tissue) allowing for more platelet binding, contributing to reduced platelet numbers in blood. Is this really pseudo-thrombocytopenia?

9. The assertion that thrombocytopenia is partially reduced upon PZQ treatment is a bit of a stretch. The is no significant difference between platelet numbers in Sm mice and Sm+PZQ mice. The assertion that thrombocytopenia is maintained after schistosomiasis infection is more in line with your data as well as perhaps even more interesting (and therapeutically relevant).

10. A key finding is that enhanced platelet clearance in Sm is associated with increased binding of platelets to Mo/Macs. You show that this is because there are more Mo/Macs binding platelets in Sm infection (Fig 4F), but also that the percentage of Mo/Macs that bind platelets increases (Supp. Fig. S4C and D). The latter point is crucial and I believe should be in the main figures not supplement.

Reviewer #3: Unclear why fecal egg counts for consecutive days were not included in figure 6. Why was relmalpha the only marker looked at? CD206/301 would be informative here in the conclusion that M2 activation is unaffected. Also why are only splenic CD4 evaluated here, liver would be informative as the authors demonstrated aggregates and extramedullary hematopoiesis in the liver

The data in Figure 7 are likely the best that can be done given the lethality of infection in the absence of platelets, however, the data reenforce the need to do the culture experiments in Figure 5 in a way that actually tests schistosome infection relevant mechanisms

PLOS authors have the option to publish the peer review history of their article (what does this mean? ). If published, this will include your full peer review and any attached files.

**Do you want your identity to be public for this peer review?** For information about this choice, including consent withdrawal, please see our Privacy Policy .

Reviewer #1: No

Reviewer #2: **Yes: ** Emma L. Houlder

Reviewer #3: No

**Figure resubmission:**

**Reproducibility:**



---

## [Decision Letter · Decision Letter 1]

14 Nov 2025

Dear Dr. Hewitson,

We are pleased to inform you that your manuscript 'Thrombocytopenia in murine schistosomiasis is associated with platelet uptake by liver macrophages that have a distinct activation phenotype' has been provisionally accepted for publication in PLOS Pathogens.

Best regards,

Constance A. M. Finney

Academic Editor

PLOS Pathogens

Margaret Phillips

Section Editor

PLOS Pathogens

Sumita Bhaduri-McIntosh

Editor-in-Chief

PLOS Pathogens

orcid.org/0000-0003-2946-9497

Michael Malim

Editor-in-Chief

PLOS Pathogens

orcid.org/0000-0002-7699-2064

Reviewer Comments (if any, and for reference):

Reviewer's Responses to Questions

**Part I - Summary**

Reviewer #1: (No Response)

Reviewer #2: The authors have fully addressed my concerns. Congratulations on your work.

**Part II – Major Issues: Key Experiments Required for Acceptance**

Reviewer #1: (No Response)

Reviewer #2: (No Response)

**Part III – Minor Issues: Editorial and Data Presentation Modifications**

Reviewer #1: The authors have addressed all of my concerns. I can now recommend the manuscript for publication.

Reviewer #2: (No Response)

PLOS authors have the option to publish the peer review history of their article (what does this mean? ). If published, this will include your full peer review and any attached files.

**Do you want your identity to be public for this peer review?** For information about this choice, including consent withdrawal, please see our Privacy Policy .

Reviewer #1: No

Reviewer #2: **Yes: ** Emma L. Houlder

---

## [Editor Report · Acceptance letter]

Dear Dr. Hewitson,

We are delighted to inform you that your manuscript, "Thrombocytopenia in murine schistosomiasis is associated with platelet uptake by liver macrophages that have a distinct activation phenotype," has been formally accepted for publication in PLOS Pathogens.

Best regards,

Sumita Bhaduri-McIntosh

Editor-in-Chief

PLOS Pathogens

orcid.org/0000-0003-2946-9497

Michael Malim

Editor-in-Chief

PLOS Pathogens

orcid.org/0000-0002-7699-2064